# Long COVID and Its Impacts: A Case–Control Study in Brazil

**DOI:** 10.3390/biomedicines13071615

**Published:** 2025-07-01

**Authors:** Cristina M. Ruas, Maria Laura Silva, Ana L. G. F. Figueiredo, Amanda P. de Alencar, Samuel de S. Melo, Geovani F. de Castro, Natália V. Carobin, Melina A. Cordeiro, Janete F. R. Aguirre, Amanda F. M. de Oliveira, Adriano de P. Sabino

**Affiliations:** 1Social Pharmacy Department, Pharmacy Faculty, Federal University of Minas Gerais, Belo Horizonte 31270-901, Brazil; geovanicastro@farmacia.ufmg.br (G.F.d.C.); janete.aguirre@gmail.com (J.F.R.A.); amandafileto25@gmail.com (A.F.M.d.O.); 2School of Pharmacy, University of Bordeaux, 33000 Bordeaux, France; maria-laura.silva@u-bordeaux.fr; 3Clinical Analysis and Toxicological Analysis, Pharmacy Faculty, Federal University of Minas Gerais, Belo Horizonte 31270-901, Brazil; analugff@gmail.com (A.L.G.F.F.); amandapacheco.alencar@gmail.com (A.P.d.A.); samuelsmelo279@gmail.com (S.d.S.M.); natyvirca@gmail.com (N.V.C.); mellinacordeiro28@gmail.com (M.A.C.); adriansabin@farmacia.ufmg.br (A.d.P.S.)

**Keywords:** post-acute COVID-19 syndrome, COVID-19, case–control studies

## Abstract

**Introduction:** Long COVID, or post-COVID-19 syndrome, refers to a set of persistent symptoms following SARS-CoV-2 infection without another identifiable cause. Studies indicate that symptoms can last for up to two years and affect multiple body systems. **Objective:** The objective of this study is to compare symptom prevalence between infected individuals pre and post-COVID-19 and non-infected individuals in a population from Southeastern Brazil. Materials and **Methods**: A case–control study was conducted with participants from the MonitoraCovid program in a university in Brazil. The study included adults who responded to a questionnaire about long COVID symptoms. Data were collected virtually between October 2023 and May 2024. **Results**: Of the 2886 individuals eligible for analysis, 75.5% reported having been positive for COVID-19. Most participants were vaccinated, with 82.99% receiving two doses. In the pre and post comparison, individuals who had COVID-19 were more likely to report increased symptoms after infection, with 95.5% of assessed conditions worsening, particularly cognitive and respiratory issues. A comparison between those who had and had not been infected with COVID-19 showed that only 6.67% of symptoms were more prevalent in the infected group. The most significant post-COVID-19 symptoms included memory problems, fatigue, and shortness of breath, though some conditions, such as anxiety and sleep disturbances, were less common among those who had COVID-19. **Conclusions**: The findings reinforce that long COVID significantly impacts cognitive health, highlighting the importance of monitoring previously infected individuals. The study also emphasizes the need for further research in Global South contexts to better understand the long-term implications of COVID-19.

## 1. Introduction

Long COVID, also known as post-COVID-19 syndrome, refers to persistent symptoms that affect individuals after the acute phase of Severe Acute Respiratory Syndrome Coronavirus 2 (SARS-CoV-2) infection without any other known cause to justify them [1]. This topic has gained relevance in public health due to its prolonged impact on individuals’ health, with studies reporting symptom persistence for up to two years after COVID-19 infection [2].

COVID-19 infection can affect a wide range of bodily systems, including respiratory, neurological, cardiovascular, gastrointestinal, and endocrine systems [3,4], leading to long-term sequelae. The World Health Organization describes the ten most common long COVID symptoms as fatigue, respiratory problems, concentration and memory issues, sleep disturbances, persistent cough, dyspnea, muscle pain, depression and anxiety, and loss of smell and taste [1].

Major studies on this topic have been conducted in the Northern Hemisphere, including Europe, Asia, and North America [3,4,5,6,7,8,9,10,11,12,13,14,15]. The most prevalent long COVID symptoms, regardless of the time interval in the post-acute phase, were fatigue and dyspnea. Most studies have focused on analyzing the prevalence of symptoms in individuals who tested positive for COVID-19, classified as pre–post studies. Few of them compared long COVID symptoms between positive and negative populations [16]. Caspersen et al. (2022) [17] followed a cohort of 70,000 adults with and without COVID-19 in Norway for up to 12 months. Based on the relative risk between previously infected and non-infected individuals, the study associated 13 of the 22 symptoms reported by participants with the SARS-CoV-2 virus [17].

The high prevalence of symptoms in individuals after COVID-19 infection is well documented; however, the difficulty in finding appropriate control groups for comparison has made it challenging to confirm that these symptoms are specific to the disease. Moreover, few studies assessing the characteristics of this population have been conducted in the Global South. In this context, we conducted a case–control study in a population residing in southeastern Brazil. The objective of this study was to compare the prevalence of self-perceived health problems among individuals who had COVID-19 and two control groups (pre–post control and a control group of individuals who were never infected with COVID-19).

## 2. Materials and Methods

### 2.1. Study Design and Participants

This is a case–control study that assesses the likelihood of individuals with long COVID symptoms having been previously exposed to the SARS-CoV-2 virus (Figure 1). The study included all participants of the MonitoraCovid diagnostic program who were monitored during the pandemic.

The MonitoraCovid program was developed by a federal university during the COVID-19 pandemic to track the disease’s progression in Brazil and provide data to support public health decisions. Utilizing geolocation technology and epidemiological intelligence, the program enabled the analysis of virus spread, assisting in identifying high-risk areas and formulating response strategies. In addition to real-time monitoring, MonitoraCovid also tracked symptoms within the academic community by conducting screenings, testing self-identified symptomatic cases at the school of pharmacy, and monitoring positive cases. It included university professionals and students, encompassing faculty members, administrative staff, undergraduate students, and outsourced workers affiliated with the institution. The academic community, comprising over 30,000 individuals, directly benefited from the program, which recorded more than 108,000 accesses, resulting in 11,645 consultations via Telecovid-19 and the confirmation of 4806 COVID-19 cases through tests conducted at the university. The program also promoted scientific dissemination and the training of healthcare professionals [18].

The program coordinators authorized access to the data from the attended community following ethical review and approval. The study population was restricted to adults aged 18 years and older. A minimum participation rate of 30% in the long COVID survey was estimated, corresponding to approximately 3483 individuals. All participants who at least answered the questions related to the COVID-19 diagnosis were considered eligible. Data collection occurred between October 2023 and May 2024 (8 months).

### 2.2. Assessment of Long COVID Using C19-YRS Tool

To assess long COVID, an adapted version of the “Modified COVID-19 Yorkshire Rehabilitation Screening (C19-YRS) Self-report” was used. The C19-YRS is a self-administered screening tool designed to define and assess long COVID, which is characterized by the persistence of symptoms for more than 12 weeks following an initial SARS-CoV-2 infection. This questionnaire addresses physical, cognitive, psychological, and functional symptoms. The C19-YRS has been recommended by the National Health Service England (NHS), and its use in this study was approved by the authors [19]. In addition to the core C19-YRS questions, the tool incorporated sections on sociodemographic information and clinical history. Adaptations were made for the inclusion of control groups (Appendix A). A pilot test was conducted with a small sample to evaluate the clarity and comprehensibility of the questionnaire. Feedback from the participants was analyzed and used to refine the final version of the online questionnaire.

The questionnaire collected data across several categories, including sociodemographic information such as age, sex, education level, skin color/race [20], and access to private health insurance. It also gathered information on clinical conditions, such as pre-existing medical conditions (e.g., heart disease and diabetes), current medication use, and COVID-19 vaccination status. Life habits were assessed, including physical activity levels, dietary habits, alcohol consumption, smoking, and illicit drug use. Participants also reported additional symptoms experienced in the past six months, such as rash, weight changes, nausea, and dizziness, as well as emotional and psychological symptoms like anxiety and depression. COVID-19 history was collected, including the number of confirmed diagnoses, the date of the last diagnosis, and hospitalization status. Lastly, the health assessment covered thirty health conditions, including shortness of breath, fatigue, cognitive impairment, pain, and sleep disturbances, with participants rating the severity of symptoms both currently (“now”) and pre-COVID-19 on a scale with classifications of mild, moderate, severe, and no change. Additionally, participants rated their current health and their health before contracting COVID-19 using a visual scale.

### 2.3. Data Collection and Statistical Analysis

Data were collected using MonkeySurvey^®^ and stored in an Excel spreadsheet. Analyses were performed using R software (R version R 4.5.1 binary for macOS Copyright (C) 2025 The R Foundation for Statistical Computing; and RStudio Copyright (C) 2025 by Posit Software, PBC). Descriptive statistics, including absolute numbers, proportions, means, and dispersion measures, were used to summarize the data. To compare mean differences, the appropriate statistical test was applied. The odds ratio (OR) was calculated to assess sociodemographic, clinical, physical, cognitive, psychological, and functional characteristics between groups of individuals who had COVID-19 (before and after infection) and those who were not infected. A test for Equal or Given Proportions (prop.test in R) was conducted to compare proportions, and Student’s *T*-test (t.test R) was performed to compare means, with statistical significance set at *p* < 0.05.

The results will be presented in the following order: sociodemographic characteristics of the participants, stratified by COVID-19 status (previously infected vs. never infected) and two analyses, namely the likelihood of symptom changes before and after COVID-19 infection and the comparison of symptom changes between those who had COVID-19 and those who did not.

### 2.4. Ethical Considerations

Information about the study was disseminated through printed materials placed in strategic locations and through the university’s internal bulletin. Personalized emails were sent to all participants from the MonitoraCovid program, inviting them to complete the questionnaire, with up to three reminder emails sent to encourage participation. To increase the response rate and achieve the target of at least 30%, follow-up messages were also sent via WhatsApp as a final attempt to gather responses. Along with the questionnaire, they received an informed consent form (ICF), detailed instructions on how to complete the survey, and a link to the official long COVID project website. The project was submitted to and approved by the Research Ethics Committee under CAAE No. 33202820.7.1001.5348.

## 3. Results

### 3.1. Sociodemographic Data

Among the 3626 respondents from the community surveyed by the MonitoraCOVID program between 2023 and 2024, 2886 were considered eligible for the study and were included in this analysis. Of those, 72.63% (n = 2096) reported a positive diagnosis test for COVID-19: 1286 (61.35%) had the disease once, 638 (30.44%) had it twice, and 169 (8.06%) had it three times or more. Regarding COVID-19 complications, 23 (1.09%) participants were hospitalized. The remaining 27.37% (n = 790) did not have a confirmed COVID-19 diagnosis according to self-reports. Of the patients who had COVID-19, 1,58% required hospitalization at least once (n = 33). Patients who did not have COVID-19 reported having suspected infection at least once (n = 416, 53.27%).

The majority of participants reported having been vaccinated: 82.99% of participants received two doses, while 17.01% received only one dose. For booster doses, 45.26% received one dose, and 50.90% received two doses. A total of 38.23% of participants received a bivalent vaccine (Table 1).

Individuals who self-reported having had COVID-19 were predominantly older (37.5 ± 12.68 years), white (62.16%), and had a postgraduate education (50.43). In contrast, individuals who reported being negative for COVID-19 were primarily self-declared as brown or black (32.94% and 11.33%, respectively), had a secondary education (31.09%), and had a higher proportion of access to private healthcare (33.33%) (Table 2).

### 3.2. Health Conditions and Lifestyle Habits

Individuals who self-reported having had COVID-19 were more likely o report continuous medication use (57.05%), to engage in regular physical activity (77.54%), to have healthy eating habits (77.66%), and consume fresh food daily (69.90%) and were less likely to smoke (85.36%) (Table 3).

### 3.3. Comparison of Health Status Before and After COVID-19 Infection

Among the 2440 individuals who self-reported having been positive for COVID-19, an increase in symptom prevalence was observed after infection compared to the pre-infection period. Among the 30 assessed health conditions, a higher prevalence of symptoms was reported in COVID-19-positive patients. Considering the 30 health conditions and the three levels of symptom intensity, we have 90 possible situations. A symptom increase was noted in 95.5% (86/90) of cases. On the health scale, the average score before infection was 90 (95%CI = 80–95), which decreased to 80 (95%CI = 70–90) after infection ([Fig biomedicines-13-01615-ch001]).

The symptoms most likely to occur in a severe form after infection with COVID-19 were difficulty finding words or understanding others (20.89), memory problems (19.06), concentration problems (10.55), planning problems (10.34), feeling unwell after emotional distress or physical efforts (7.19), palpitations (5.88), difficulty in broad daily activities (4.85), sleep disturbances (4.81), anxiety (4.40), having unwanted memories (4.08), dizziness (3.96), difficulty walking or moving (3.90), abdominal pain (3.65), fatigue in usual activities (3.42), trying to avoid thoughts (3.28), depression (3.20), socialization or interaction problems (3.00), shortness of breath when climbing stairs (2.77), headache (1.81), and cough and sensitivity (1.76) ([Fig biomedicines-13-01615-ch001]).

The symptoms most likely to occur in a moderate form were memory problems (11.64), difficulty finding words or understanding others (9.20), concentration problems (6.98), planning problems (4.75), fatigue in usual activities (4.70), dizziness (4.61), palpitations (4.51), feel unwell after emotional distress or physical efforts (3.91), difficulty in broad daily activities (3.48), socialization or interaction problems (3.31), shortness of breath when climbing stairs (3.26), abdominal pain (3.20), sleep disturbances (3.14), joint pain (2.93), trying to avoid thoughts (2.85), having unwanted memories (2.76), chest pain (2.68), anxiety (2.41), difficulty walking or moving (2.33), cough and sensitivity (2.31), muscle pain (2.23), depression (2.16), voice changes (2.10), altered sense of smell (1.74), headache (1.67), shortness of breath when getting dressed (1.13), and shortness of breath when changing position (1.01) ([Fig biomedicines-13-01615-ch001]).

Finally, the symptoms most likely to occur in a mild form were having unwanted memories (3.37), memory problems (2.76), dizziness (2.75), difficulty finding words or understanding others (2.68), palpitations (2.53), altered sense of smell (2.50), difficulty in broad daily activities (2.38), trying to avoid thoughts (2.37), shortness of breath when getting dressed (2.28), having unpleasant dreams (2.25), altered sense of taste (2.25), feel unwell after emotional distress or physical efforts (2.10), concentration problems (2.09), chest pain (2.05), planning problems (1.99), cough and sensitivity (1.98), shortness of breath when changing position (1.98), fatigue in usual activities (1.95), difficulty walking or moving (1.83), shortness of breath when climbing stairs (1.79), voice changes (1.73), shortness of breath at rest (1.70), joint pain (1.68), abdominal pain (1.53), muscle pain (1.53), socialization or interaction problems (1.44), sleep disturbances (1.32), headache (1.25), depression (1.24), and anxiety (1.23) ([Fig biomedicines-13-01615-ch001]).

### 3.4. Comparison of Health Status Between Individuals Who Had and Did Not Have COVID-19

We compared the health status of individuals who had COVID-19 (n = 2440) with those who did not (n = 790). Considering the 30 health conditions and the three levels of symptom intensity, we have 90 possible situations. A higher prevalence of symptoms was observed only in 6.67% (6/90) ([Fig biomedicines-13-01615-ch001]).

The symptoms most likely to occur in a severe form in the COVID-19 group were feeling unwell after emotional distress or physical effort (OR = 2.11), memory problems (OR = 1.59), and planning problems (OR = 1.17). Those most likely to occur in a moderate form were altered sense of smell (OR = 1.83), memory problems (OR = 1.45), and planning problems (OR = 1.06). The symptom most likely to occur in a mild form was shortness of breath when getting dressed (OR = 1.60) ([Fig biomedicines-13-01615-ch001]).

In contrast, in 23 out of the 90 assessed conditions (25.55%), a lower likelihood of occurrence in the COVID-19 group was detected. The symptoms with lower prevalence among individuals with COVID-19 were sleep disturbances (OR = 0.71), anxiety (OR = 0.46), trying to avoid thoughts (OR = 0.03), having unpleasant dreams (OR = 0.04), and having unwanted memories (OR = 0.05) (severe); those with moderate presence were anxiety (OR = 0.56), having unpleasant dreams (OR = 0.01), trying to avoid thoughts (OR = 0.02), and having unwanted memories (OR = 0.05); and those with mild prevalence were planning problems (OR = 0.89), joint pain (OR = 0.80), abdominal pain (OR = 0.78) depression (OR = 0.75), cough and sensitivity (OR = 0.71), socialization or interaction problems (OR = 0.70), sleep disturbances (OR = 0.77), headache (OR = 0.66), muscle pain (OR = 0.62), voice changes (OR = 0.58), anxiety (OR = 0.53), trying to avoid thoughts OR = (0.04), having unpleasant dreams (OR = 0.04), and having unwanted memories (OR = 0.08) ([Fig biomedicines-13-01615-ch001]).

## 4. Discussion

This research represents progression regarding the existing scientific literature as it analyzes the likelihood of symptom escalation before and after COVID-19 infection and compares individuals who have self-declared having been positive or negative for the disease.

In the initial comparison, a rise in health issues following infection was noted in 95.5% of the examined cases. These included cognitive challenges, fatigue, heart palpitations, sleep disturbances, anxiety, depression, muscle aches, and difficulties with memory, focus, and planning. Other symptoms included dizziness, abdominal discomfort, shortness of breath, alterations in smell and taste, joint pain, coughing, and heightened sensitivity.

The comparison between self-reported infected and uninfected individuals showed that only 6.67% of health issues were more common among the infected individuals. Conditions that persisted in those infected with COVID-19 included feeling ill following emotional stress or physical exertion, difficulties with memory and planning, a diminished sense of smell, and breathlessness while dressing. Interestingly, 25.55% of symptoms, including anxiety, sleep disturbances, and muscle soreness, were less frequently reported by the infected group.

Our findings indicate that COVID-19 has had enduring effects on well-being, particularly mental health. However, many symptoms attributed to the disease may be linked to the socio-economic challenges stemming from the health and political crises of that period [21]. The pandemic has exacerbated health inequalities worldwide, disproportionately affecting individuals from lower socio-economic backgrounds who face barriers to essential healthcare services [22]. Additionally, COVID-19 triggered global economic and social crises, impacting employment, income security, informal workers, gender disparities, and mental health. These disruptions, along with restrictions on economic activities and widespread financial instability, have contributed to heightened fear and anxiety [23,24].

This study reported symptoms associated with the neurological and cognitive systems, including fatigue, issues with concentration and memory, difficulties in planning, challenges in word retrieval or comprehension, and fatigue following physical exertion. The human body typically performs cognitive tasks like understanding language, planning, remembering, managing energy, and concentrating. These abilities are supported by brain flexibility that aids learning and adjustment, contributes to proper blood flow in the brain, prevents nerve damage, helps maintain low inflammation in the body, and helps maintain a good supply of oxygen, brain chemicals, and sugar. Ongoing inflammation [25] observed in cases of long COVID here and in other studies may be linked to the emergence of these symptoms [25].

Research on long COVID frequently identifies significant issues with concentration and memory, with prevalence rates of 75% and 78%, respectively [26]. A study conducted in the United States, involving 435 patients who had previously tested positive for SARS-CoV-2, representing 6% of the 7305 identified cases, investigated long COVID recovery. The study found a high prevalence of subjective cognitive decline among the long COVID group, with 61.8% of participants reporting cognitive deficits compared to 29.1% of patients without long COVID [27].

Another study assessed the physical and mental health of adolescents who had previously tested positive for SARS-CoV-2 in England three months after SARS-CoV-2 infection. A significant proportion reported persistent symptoms, including fatigue and emotional difficulties [28]. Additionally, a longitudinal study investigated the impact of the pandemic on the mental health of individuals in the Netherlands. They found that the pandemic exacerbated anxiety and depression symptoms, particularly among individuals with pre-existing psychiatric disorders [29]. Østergaard et al. (2021) [30] suggested that alterations in capillary transit time may contribute to persistent long COVID symptoms, such as fatigue and brain fog. These findings indicate that therapeutic approaches aimed at improving capillary circulation may benefit patients experiencing prolonged post-infection sequelae [30]. A comprehensive review of the main findings on long COVID was conducted, including the underlying mechanisms, persistent symptoms, and public health implications. The study suggests that symptom persistence is associated with chronic inflammation, immune dysfunction, and vascular damage [31].

Anosmia and dysgeusia, frequently reported in COVID-19 cases, result from viral damage to sensory cells and the olfactory epithelium. The SARS-CoV-2 spike protein targets ACE2-expressing sustentacular and taste bud epithelial cells, leading to inflammation and impaired sensory function [32]. As flavor perception involves both taste and retronasal olfaction, these symptoms are closely interconnected. The frequency at which modifications in the sense of smell, also known as olfaction (which occurs in approximately 48% of individuals), as well as changes in the sense of taste (observed in about 42% of people), has been thoroughly recorded and analyzed in various scholarly studies that focus on the ongoing and complex condition referred to as long COVID [4]. Boscolo-Rizzo et al. (2021) found that after 12 months, 21.3% of patients reported persistent COVID-19-associated chemosensory dysfunction, with 69.5% experiencing complete recovery, 21.9% reporting partial improvement, and 8.6% maintaining or worsening symptoms [33]. Another study also identified that after 12 months, patients reported persistent COVID-19-associated chemosensory dysfunction. This dysfunction, which affects the sense of taste and smell, can significantly impact the quality of life and complicate recovery efforts for those affected by long COVID [34].

Shortness of breath in COVID-19 results from lung inflammation, which impairs alveolar function and gas exchange. Infection of type II alveolar cells reduces surfactant production, leading to alveolar collapse and decreased oxygenation [35]. Pulmonary fibrosis further compromises lung elasticity, causing persistent dyspnoea, even at rest [36]. Difficulty breathing during movement reflects the body’s struggle to regulate ventilation and circulation efficiently. Cortes-Telles et al. (2021) highlight that there is a true physiological mechanism that may explain persistent dyspnea after COVID-19 [37]. A study conducted with post-COVID-19 patients found that five months after discharge, 51% reported persistent dyspnea [38]. Another study estimated that COVID-19 survivors have twice the risk of developing respiratory conditions [39].

Our results are consistent with previous studies. A systematic review of 8591 COVID-19 survivors found that one year post-infection, the most common symptoms included fatigue/weakness (28%), dyspnea (18%), arthromyalgia (26%), depression (23%), anxiety (22%), memory loss (19%), and concentration difficulties (18%) [10]. Similarly, a prospective cohort study in Brazil revealed that over one-third of patients experienced persistent symptoms, particularly dyspnea, fatigue, and daily activity limitations, regardless of the initial disease severity [40]. Additionally, one in five COVID-19 survivors aged 18–64 and one in four survivors aged ≥65 experienced at least one new health condition attributable to the infection [39]. These findings reinforce the long-term burden of COVID-19 and the need for continued monitoring and support for affected individuals.

In summary, the findings underscore the complex and persistent nature of long COVID, which affects various organ systems and significantly impacts patients’ quality of life. From a neurological perspective, severe symptoms such as memory problems and planning difficulties, along with moderate cognitive impairments, highlight the enduring cognitive effects of the condition. Additionally, the respiratory system is notably affected, with mild symptoms like shortness of breath being observed during routine activities such as dressing. The impact on the autonomic nervous system is reflected in the severe symptom of feeling unwell after emotional distress or physical effort (Figure 2). While some individuals recover completely, a substantial proportion continue to experience debilitating symptoms, necessitating targeted clinical guidelines and therapeutic interventions. Furthermore, longitudinal studies are crucial to elucidate the underlying mechanisms, identify at-risk populations, and develop effective treatment strategies. Addressing the long-term consequences of COVID-19 requires an integrated approach combining biomedical research, patient-centered care, and public health policies to mitigate the burden of this emerging condition.

This case–control study presents some limitations inherent to the methodological design [41]. A major limitation of this study is the use of self-reported COVID-19 diagnosis, which may introduce classification bias. Individuals who did not undergo diagnostic testing for every episode of respiratory symptoms during the pandemic may be unable to accurately confirm that they never had the infection, potentially leading to the underreporting of cases. Nevertheless, self-reporting can be advantageous as it allows for the inclusion of diagnoses obtained from different laboratories and pharmacies, expanding the collection of cases that might not be recorded in MonitoraCovid.

Another limitation relates to the representativeness of the sample as the study was conducted in an academic institution, restricting participation to adults affiliated with the university, which may limit the generalizability of the findings to the broader population. However, this study also has important strengths. Notably, the use of two distinct control groups (pre–post and individuals who were never infected) strengthens comparisons and reduces confounding biases [41]. Additionally, the sample consisted of participants who were previously monitored in the MonitoraCovid program, ensuring access to longitudinal data and detailed health histories, a distinguishing factor compared to other long COVID studies. The adoption of a validated instrument (C19-YRS) for symptom assessment further enhances the reliability of the findings, reinforcing the methodological robustness of the study. Finally, the inclusion of a significant number of participants from the Global South, where long COVID studies are scarce, represents an important scientific contribution.

## 5. Conclusions

This study significantly contributes to the understanding of the long-term effects of COVID-19 infection on individuals’ health by comparing the prevalence of symptoms among those who had the disease and two control groups. The results indicate an increase in symptom occurrence after infection. Otherwise, the comparative analysis between infected and non-infected individuals demonstrated that only a small proportion of symptoms were significantly more prevalent among those who had COVID-19, particularly in cognitive and psychological domains, such as difficulties with concentration, memory, and planning; fatigue; breathing difficulties; and altered taste perception. It is suggested that other socio-economic variables may influence the perception of reported symptoms.

The findings reinforce the importance of monitoring individuals with a history of COVID-19, especially concerning mental health and cognition, and highlight the need for further studies to distinguish the direct impacts of infection from the indirect consequences of the pandemic. Additionally, this study expands the literature on long COVID in a Global South context, providing an essential perspective for planning public health strategies tailored to this reality.

Given the substantial impact of infection on individual well-being, it is essential to implement health policies that consider both the rehabilitation of patients affected by long COVID and preventive measures to minimize the burden of this condition on the population. Future studies should explore underlying pathophysiological mechanisms and socioeconomic factors that may influence symptom persistence, ensuring a more integrated and effective approach to managing long COVID.

## Data Availability

The data will be made available upon request to the corresponding author.

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
