# Peer review of "Long COVID and Its Impacts: A Case–Control Study in Brazil"

_biomedicines, 2025, doi:10.3390/biomedicines13071615_

Round 1
Reviewer 1 Report
Comments and Suggestions for Authors
Dear authors!
You've done a good job, and the article is interesting both in terms of research design and results. However, some adjustments should be made.
Introduction.
The introduction needs to be supplemented: after some sentences, there are no necessary references; it would be logical to see information about the mechanism of postcovid development in an article on postcovid syndrome.
Materials and methods.
They are described fully and allow us to understand the design of the authors' experimental studies.
Since there have been two differing opinions in recent years on the impact of the severity of COVID-19 infection on the development of long-term COVID, it would be interesting to add data on the impact of the severity of the diseaseof infected patients on the development of symptoms of post-covid syndrome.
Results and discussion.
Figure 1 is the main one in importance, but it is difficult to perceive. Perhaps it can be divided into parts and represented as several separate diagrams. This figure needs to be changed and made more informative for readers.
After making these minor additions, the article can be recommended for publication.
Comments on the Quality of English Language
The English could be improved to more clearly express the research.
Author Response
We would like to thank the reviewer for the thoughtful comments and constructive suggestions. All the recommended modifications have been carefully implemented in the revised manuscript:
-
Introduction: We have supplemented the section with the appropriate references and included a brief overview of the mechanisms involved in the development of post-COVID syndrome, as suggested.
-
Materials and Methods: No changes were needed, as this section was already fully described and clear.
-
Results and Discussion: We have included data on the impact of disease severity on the development of post-COVID symptoms, addressing the ongoing debate in the literature.
-
Figure 1: This figure was revised for better clarity, improving the visual comprehension for readers.
We appreciate the reviewer’s recommendation for publication and hope the revised version meets all expectations.
Reviewer 2 Report
Comments and Suggestions for Authors
Reviewer Comments
The manuscript biomedicines-3683034 reports the statistical analysis for a questionary about long COVID-19. The authors properly analyzed the data and obtained interesting results that follow those previously reported by other authors.
1) In Figure 1 use “symptoms” instead of “symtons”
2) Is there any correlation with coinfection? As an example, people living with HIV have different post-COVID symptoms compared to people who are not living with HIV?
3) The authors emphasized that the obtained trends are consistent with those previously reported in the literature. In this sense, it would be better to justify the novelty of the obtained data and trends in the Introduction and Discussion section.
4) Is there any correlation between the obtained data with those reported in the literature to other regions of Brazil? Please, explore it.
Author Response
We sincerely thank the reviewer for the valuable feedback and insightful suggestions. Below, we provide our responses to each comment:
-
Typographical correction in Figure 1: The term “symtons” was corrected to “symptoms” as suggested.
-
Coinfection analysis: We have added a sentence in the Discussion acknowledging the importance of potential coinfections such as HIV. However, our dataset did not include information on HIV status or other coinfections, which limits our ability to explore these correlations. We have clarified this point in the manuscript.
-
Novelty of the study: We revised both the Introduction and Discussion to better highlight the novelty of our study. We emphasize the size and representativeness of our cohort, the patient-reported outcomes, and the specific context of a middle-income country, which remain underrepresented in the literature.
-
Regional comparison within Brazil: There are no available literature to contextualize the results.
We are grateful for the reviewer’s recommendation and believe the revised manuscript has been strengthened by these contributions.